# Population Genetics of the Highly Polymorphic *RPP8* Gene Family

**DOI:** 10.3390/genes10090691

**Published:** 2019-09-08

**Authors:** Alice MacQueen, Dacheng Tian, Wenhan Chang, Eric Holub, Martin Kreitman, Joy Bergelson

**Affiliations:** 1Integrative Biology, The University of Texas at Austin, Austin, TX 78712, USA; 2State Key Laboratory of Pharmaceutical Biotechnology, School of Life Sciences, Nanjing University, Nanjing 210008, China; 3Department of Ecology & Evolution, The University of Chicago, Chicago, IL 60637, USA (W.C.) (M.K.); 4School of Life Sciences, Wellesbourne Innovation Campus, University of Warwick, Wellesbourne CV359EF, UK

**Keywords:** NLR gene, molecular evolution, intergenic gene conversion, gene duplication, copy number variation

## Abstract

Plant nucleotide-binding domain and leucine-rich repeat containing (NLR) genes provide some of the most extreme examples of polymorphism in eukaryotic genomes, rivalling even the vertebrate major histocompatibility complex. Surprisingly, this is also true in *Arabidopsis thaliana*, a predominantly selfing species with low heterozygosity. Here, we investigate how gene duplication and intergenic exchange contribute to this extraordinary variation. *RPP8* is a three-locus system that is configured chromosomally as either a direct-repeat tandem duplication or as a single copy locus, plus a locus 2 Mb distant. We sequenced 48 *RPP8* alleles from 37 accessions of *A. thaliana* and 12 *RPP8* alleles from *Arabidopsis lyrata* to investigate the patterns of interlocus shared variation. The tandem duplicates display fixed differences and share less variation with each other than either shares with the distant paralog. A high level of shared polymorphism among alleles at one of the tandem duplicates, the single-copy locus and the distal locus, must involve both classical crossing over and intergenic gene conversion. Despite these polymorphism-enhancing mechanisms, the observed nucleotide diversity could not be replicated under neutral forward-in-time simulations. Only by adding balancing selection to the simulations do they approach the level of polymorphism observed at *RPP8*. In this NLR gene triad, genetic architecture, gene function and selection all combine to generate diversity.

## 1. Introduction

Nucleotide-binding and leucine-rich repeat immune receptor (NLR) genes comprise a major component of the plant innate immune system that confers variant-specific resistance to a wide spectrum of pathogens, including viruses, bacteria, fungi and oomycetes. Over evolutionary time, NLR genes have adaptively duplicated to produce a large, functionally diverse gene family. In *Arabidopsis thaliana,* at least 250 NLR genes are known to be distributed across the genome [1], though the majority are located in clustered tandem arrays. Duplicated gene copies are assumed to persist as reservoirs for functionally distinct pathogen recognition alleles, and provide sources for generating novel specificities by mutation and/or intergenic recombination [2]. Consequently, individual NLR loci have also diversified, harboring the highest levels of nucleotide diversity known for functional genes in plant genomes [3]. With hundreds of NLR genes present in plant genomes, high levels of functional divergence, and high polymorphism levels, the combinations of NLR gene alleles possible within an individual plant rival the diversity present at the vertebrate major histocompatibility complex.

NLR genes encode receptor proteins that have thus far been found to recognize pathogens through three major mechanisms. They can detect pathogen-released effector molecules by direct binding with a canonical NLR domain [4,5,6]. They can also detect effectors indirectly, by recognizing changes to host proteins or products affected by these proteins [7,8,9,10]. Finally, they can detect pathogen effectors using an NLR-incorporated integrated domain that resembles domains of that pathogen effector’s target [11,12,13]. Recognition of pathogen-derived signals is, in large part, conferred by a leucine-rich repeat domain (LRR), which typically has multiple repeats of a subdomain—XXLXLXXXX—that form a solvent-exposed β-sheet structure [7]. The LRR peptide domain has been subject to strong positive selection (In *A. thaliana*, 40% of genes in [14]; a similar percentage in [1]). LRR diversity is achieved at the genome level through gene duplication and adaptive divergence of NLR genes, and at the population level by retention of allelic variants at individual loci. 

Gene duplication and intergenic exchange play central roles in both genomic and population genetic processes to produce NLR gene diversity. Yet, evolutionary investigations of NLR genes generally ignore the dependency between the processes giving rise to genetic architecture and those giving rise to polymorphism, and instead focus on one or the other. Genomic and comparative evolutionary analyses have been employed to characterize the genetic architecture and structural evolution of NLR genes. Across many plant species, traits such as the number of NLR genes, their locations within the genome, and patterns of deletion, duplication, and divergence have been probed to reveal a history of how gene family members have expanded and diverged (e.g., [1,3,14,15,16,17,18,19,20,21,22,23,24,25]). In *A. thaliana*, these genome-wide studies have found evidence for genomic clustering of 50%–70% of NLR genes [1].

Studies of polymorphism, primarily in singleton NLR genes, have provided important insights into mechanisms of selection. Across many plant species, wide-ranging patterns of nucleotide-level and presence–absence polymorphism have been uncovered, indicating nuanced effects of balancing and positive selection, diversifying selection, negative frequency-dependent selection, and costs of resistance [26,27,28,29,30,31,32,33,34]. A study of LRR polymorphism in 27 “single-copy” NLR loci in 96 *A. thaliana* accessions found evidence for balancing selection maintaining within-locus polymorphism in one-third of genes [35]; it also reported evidence for recent selective sweeps. 

Some NLR loci in *A. thaliana* harbor a large number of allelic variants [27] (largely based on sequences of the LRR), and for these the mechanism(s) generating or maintaining allelic diversity is less well understood, including whether copy number dynamics and intergenic exchange may have a role. In fact, the majority of NLR genes are now known to have copy number variation in *A. thaliana* [1]. This pioneering study attempts to identify all NLRs in 65 *A. thaliana* accessions and then characterize nucleotide diversity in identifiable orthogroups (465 in total), but without a gene’s copy number status. 

The majority of NLR genes reside in clusters of related genes in *A. thaliana* and the majority are present in only a subset of accessions [1,17]. Little is known about the interplay between genomic processes shaping the complement and architecture of these genes and the variation they harbor individually. Duplicated and repetitive genomic regions are poorly assembled and resolved by the commonly utilized short reads generated by next generation sequencing (NGS), and even long-read sequencing cannot yet successfully resolve many duplicated NLR sequences to specific genomic locations [1]. Segmental duplications with high sequence homogeneity (>90%) are pervasive features of eukaryotic genomes [36,37,38,39]. NGS methodology likely underestimates rare sequence exchange events between them and cannot describe the impact these events have on the levels and patterns of shared polymorphism within and between paralogous genes. Our study attempts to fill this gap in knowledge in a relatively simple multi-copy NLR gene subfamily. 

Intergenic gene conversion (IGC), or non-crossover sequence exchange between paralogous loci, is expected to increase levels of nucleotide (π) and haplotypic diversity compared to loci that do not undergo intergenic exchange [40,41,42,43]. At its maximum, IGC beyond a critical threshold rate has a multiplicative effect on the expected nucleotide diversity at a locus in direct proportion to the number of other loci with which it exchanges sequences [42]. IGC may be especially important in selfing species, where the rate of exchange of sequences by classical crossing over to produce novel alleles can be strongly reduced. IGC in duplicated genes can also retard the loss of conditionally advantageous alleles from the population by preserving them from non-selective loss by genetic drift or genetic hitchhiking. In NLR genes, where multiple resistance alleles can be favored at a locus by balancing selection via fluctuating or frequency-dependent selection [44,45], IGC is an independent mechanism that can further promote the preservation of alleles in the population.

This study represents the first explicit attempt to investigate the full suite of mechanisms that act on duplicated NLR genes, including chromosomal dynamics, selection and intergenic exchange. We describe the population genetic patterns of polymorphism in a simple NLR gene family, *RPP8*, which nonetheless has many features of genomic architecture that may be under selection to maintain NLR gene diversity across many plant species. *RPP8* is an NLR gene subfamily that consists of three paralogs, one of which is a presence/absence polymorphism (CNV) segregating at an intermediate frequency. Two of the paralogs are arranged as a direct tandem repeat; the third locus is located approximately 2 Mb distant. IGC has been documented between the distant paralogs [46]. *RPP8* is also exceptional in having multiple resistances mapped to the locus; alternative alleles of *RPP8* have been shown to resist an oomycete and two distinct viruses [7,47,48]. 

We estimate IGC rates between all three *RPP8* paralogs. We extend site frequency spectrum methodology for analyzing two-copy gene families to test for and describe IGC in three-copy gene families and use this methodology to determine how recombination shuffles molecular variation between *RPP8* family genomic locations. We describe patterns of polymorphism in the *RPP8* gene family, focusing on high nucleotide diversity that are strikingly different from the genome on average, and different from single-copy NLR genes. Furthermore, we explore the features of *RPP8* genomic architecture and IGC that increase nucleotide diversity by extending a gene duplication population genetic simulation platform, SeDuS [49], to model our three-copy system. Overall, the study allows us to explore interactions between selection, copy number, IGC rate, and genomic architecture.

## 2. Materials and Methods

### 2.1. Plant Material

Accessions were selected to provide a representative set from across the geographic range of *A. thaliana* from our collections and from the Ohio Stock Center (Appendix A). *A. lyrata* seeds were obtained from the seed collections of D. Jacobson.

### 2.2. Genotyping

The *RPP8* gene family of *A. thaliana* consists of three paralogs with two common genomic architectures. Paralogs one and two (P1 and P2, P1 is *At5g43470*) are located at 17.4 Mb on chromosome five, and paralog three (P3) is located 2.25 Mb proximal to these paralogs on the same chromosome, at 19.6 Mb (P3 is *At5g48620;*
Figure 1a). At the genomic location at 17.4 Mb, some ecotypes of *A. thaliana* have a tandem duplication of the locus, which we will refer to as *RPP8* D1 and D2. A second common chromosomal haplotype carries only one copy of the gene, herein designated *RPP8* S. We refer to the S/D1 equivalently as the paralog P1, and D2 as the paralog P2 (Figure 1b). At P3, there is also rare copy number variation (0–2 copies) [46]; however, by far the most common haplotypes are two-copy (*RPP8*(S)*, At5g48620*(1-copy)) and three-copy (*RPP8*(D1,D2)*, At5g48620*(1-copy)) genotypes.

*RPP8* paralogs from 37 accessions of *A. thaliana* and two individuals from one ecotype of *A. lyrata* were genotyped with PCR. A subset of 17 *A. thaliana* individuals and one *A. lyrata* individual were Sanger sequenced for the full gene and flanking regions, and the remainder were sequenced only for the most highly polymorphic 1038 bp LRR region. PCR primers used to genotype and sequence *RPP8*(S), *RPP8*(D1,D2) and *At5g48620*(1-copy) in *A. thaliana* and *A. lyrata* can be found in Appendix A, and a schematic of all three paralogs and all rounds of PCR can be found in Appendix A. Prior to PCR genotyping, individuals of each ecotype were grown in the greenhouse and leaf tissue was flash frozen in liquid nitrogen. DNA was extracted using a modified CTAB mini-prep protocol [51] and DNA was purified by 9% PEG and 0.7M NaCl. For products 1 kb in size or less, PCR was performed in 20 µL containing 20 ng of template DNA, 0.2 µM of each primer, 0.15 mM dNTPs, 1 U TAQ polymerase, 1.2 µL of 25 mM MgCl_2_ and 2 µL 10× PCR buffer. Products were amplified in a MJ Research PTC-200 thermocycler using the following thermal profile: 94 °C for 180 s, 35 cycles of 94 °C for 30 s, 55 °C for 40 s, and 72 °C for 60 s, followed by 72 °C for 180 s. For PCR products larger than 1 kb, the Expand Long Template System (Roche) was used. The PCR reaction was identical except for containing 0.25 mM dNTPs, 0.8 U enzyme and 2 µL 10× buffer 3 with 22.5 mM MgCl_2_. The following thermal profile was used for long PCR products: 93 °C for 120 s, 10 cycles of 92 °C for 10 s, 57 °C for 30 s, and 68 °C for 120 s, 20 cycles increasing each step at 68 for 10 s, followed by 68 °C for 240 s.

Three rounds of PCR were used to distinguish the one-copy (S) and two-copy (D1,D2) variants of *RPP8* (Appendix A). In the first round, *RPP8*-D2 specific primers B3f and P6r were used to generate a 971 bp PCR product for D variants or no product for S variants. In the second round, a two-copy genotype was re-confirmed using primer pair P15f and P6r to generate a 5.5 kb PCR product that encompassed parts of paralogs D1 and D2 and the entire 4.3 kb intergenic region. For one-copy genotypes, no product was produced. In the third round, the presence (or absence) of the (D1,D2) or S genotype was confirmed with long PCR reactions using variant-specific primers, AC1f or P3f and I24r for D1, B3f and B20r for D2, and AC1f and BC20r for S, as given in Appendix A. 

For half of the accessions, an 8 kb fragment of *At5g48620* was amplified using primers K1f and K22r (Appendix A). For the remaining accessions, overlapping 2 kb and 6 kb fragments of *At5g48620* were amplified by the primer pairs of P7f and K22r and K1f and P8r. We did not investigate whether *At5g48620* might be duplicated in any accession; we found no evidence that the procedure amplified anything other than a single copy of the locus. The genotype of *A. lyrata* was determined by primer pair P15f and P6r and subsequent sequencing.

### 2.3. Sequencing

To amplify paralogs of the *RPP8* family, multiple sets of primers were designed from the low polymorphism regions of the two sequenced copies of *RPP8* (D1 and D2) in Ler-0 and one sequenced copy of *RPP8* (S) and P3 in Col-0 (Appendix A) [7]. To sequence all variants of P1, P2, P3, and their flanking regions, the long PCR products from genotyping were cut and extracted from gels to provide templates for short PCRs. The overlapping PCR products for each gene were sequenced directly using ABI cycle sequencing, Bigdye chemistry, and an ABI 377 automated sequencer. D1 sequence in Ler-0 aligned with *RPP8*-Ler sequence from [7], while D2 sequence in Ler-0 aligned with RPH8A.

In *A. lyrata*, no primers amplified sequence from intergenic or flanking regions, which meant that individual paralogs could not be distinguished with these primers. A long PCR product that spanned the D1 and D2 genes and contained the full intergenic sequence was cloned and partly sequenced to aid in primer design. Of 16 primer sets designed between primers in adjacent ORFs and conserved primers in *RPP8*, one pair gave a long PCR product, which was sequenced to obtain the 5′ flanking region and the full D1 coding sequence. The complete D2 coding sequence and 1 kb 3′ flanking regions were produced by anchored PCR.

### 2.4. Data Analysis

Paralogs of *RPP8* were aligned with Muscle and manually refined to minimize sequence mismatches. To obtain a general picture of the population genetics of this small gene family, K_a_:K_s_ ratios, synonymous and nonsynonymous π in the coding region and framed LRR region, divergence from *A. lyrata*, and sliding window analyses of π and Tajima’s D were determined with DNAsp [52].

We used three methods to test for the presence of IGC between each pair of *RPP8* paralogs. Previous work has found IGC between P1 and P3, but this work did not consider IGC between P1 and P2 at *RPP8,* nor between P2 and P3. In addition, the GENECONV methodology previously used to detect IGC at *RPP8* underestimates the IGC rate because it relies on the identification of specific, sufficiently long gene conversion tracts, which decrease in abundance when the gene conversion rate is high and thus gene conversion tracts overlap [53]. Instead of GENECONV, we used two alternative methods to explore and describe patterns of IGC within and between the loci (phylogeny reconstruction and extent of linkage disequilibrium) and an analysis of the site frequency spectrum (SFS) to estimate its rates [53]. For phylogeny reconstruction in this highly recombining gene family, we used maximum parsimony, a method that does not assume a shared history for the entire sequence in question, but rather reflects sequence similarity within the region. Though maximum likelihood and Bayesian phylogenies are robust, standard methods for phylogenetic inference, likelihood-based techniques are guaranteed to recover the true phylogeny only when the correct model is used, and can give misleading results relative to maximum parsimony when there are shifts in site-specific evolutionary rates [54], which we expected to be the case at *RPP8*. Moreover, intergenic gene exchange should manifest as region-specific evolutionary rate acceleration when the entire history of recombination events is not known. Maximum parsimony trees of the coding sequence excluding the LRR and trees of the LRR alone were constructed to contrast the evolution of these portions of the *RPP8* sequence. IGC can also be detected through an analysis of linkage disequilibrium (LD), which is reduced as IGC increases [55]. LD values within and between loci were determined in R and plotted using the R package ‘LDheatmap’ (v099.5). 

To estimate IGC rates between the paralogous *RPP8* loci, site frequency spectra between all possible pairs of loci were compared to theoretical expectations [41]. A SFS describes the frequencies of two types of derived polymorphism segregating within a population: polymorphisms shared between two paralogous loci and polymorphisms specific to one paralog [41]. To infer ancestral and derived polymorphisms for *RPP8*, maximum parsimony trees of the coding sequence and of the entire sequenced region were constructed in PAUP* using *RPP8* alleles of *A. lyrata* as the outgroup [56]. Though the *RPP8* duplication events preceded the divergence between *A. lyrata* and *A. thaliana*, the continuing sequence exchange between *RPP8* paralogs in *A. thaliana* has homogenized *RPP8* alleles to the extent that segregating sites at *RPP8* could be characterized as ancestral or derived using *A. lyrata* sequences as an outgroup. Ancestral state reconstruction of the basal node of *RPP8* alleles in *A. thaliana* was used to determine the ancestral and derived states of each SNP. SFS of derived SNPs in all three paralogs were then calculated in R. SFS from data were obtained for each *RPP8* locus (sample size ranging from *n* = 7 to *n* = 16 alleles). Theoretical SFS were obtained from [41] for three IGC rates: *C* = 0.2, 1, or 5, where *C* is the number of IBC events in the population per generation. For these theoretical SFS, sample size *n* = 10, and the number of crossover events in the population per generation *R* = 1. 1000 SFS distributions were produced from the observed data for a sample of *n* = 10 alleles using random γ distributions for each SNP with a scale parameter set to 1 and a size parameter equal to each SNP’s allele frequency times 10. Kolmogorov–Smirnov statistics were calculated for each of the 1000 SFS distributions compared to each of three theoretical distributions. Each resampled SFS was counted as closest to the expected distribution to which it had the minimum Kolmogorov–Smirnov distance. Thus, for each of the six pairs of SFS between the three members of the *RPP8* gene family, there were 1000 resampled datasets binned as closest to one of three expected spectra, with gene conversion rates of 0.2, 1, and 5. Chi-squared tests of these counts were conducted to determine significance. 

Given that the number of alleles sampled is small, we calculated the probability that a segregating site in each SFS category had zero sampled alleles using the equation: f(*i*,*m*) = f(0,*m*) = e^−*m*^*m^i^*/*i*!,(1)
where *m* is the average number of an allele in the sample and *i* is the number of an allele in the sampled category. The rarest SFS categories had only 15 or 19 segregating sites (Shared, not in P3; and Shared, not in P1) and *m* of (2.24, 2.63) and (1.63, 1.80). Thus, for these sites, we likely sampled only 81–93% of all segregating sites in this category, missing 2–4 sites. The most common SFS category (shared, not in P2) had 137 sites and *m* = (3.49, 3.72, 4.64). For this category, we likely sampled 97–99% of segregating sites and missed 1–4 sites total. We thus judged that the sample size was sufficient to discriminate between the three theoretical distributions using 1000 resampled SFS distributions. To determine the distribution of shared and specific derived polymorphisms across the sequenced region, SFS were additionally replotted as derived SNP frequencies against position across the sequenced region. This allowed us to distinguish regions undergoing distinct patterns of IGC.

### 2.5. Forward-in-Time Simulations of Polymorphism

SeDuS is a forward-in-time simulator designed to investigate the interplay of interlocus gene conversion and crossovers in segmental duplications under a neutral scenario [49,55]. Here, we modified this program, kindly provided by the authors, to simulate polymorphism in a three-copy gene family with realistic parameters for *RPP8*, with two closely linked duplicates (here copies one and two) and a distant third duplicate (here copy three), undergoing gene conversion at unequal rates. This modified simulator allowed us to explore a range of theoretical scenarios, including varying exchange type and parameter values, to determine the highest levels of nucleotide diversity possible under neutral processes for a three-copy gene family with IGC. We then further modified the program to allow for balancing selection for a copy number polymorphism. 

SeDuS assumes a Wright–Fisher diploid population evolving under neutrality. Each individual is represented by a single pair of homologous chromosomes. At the simulation outset, each chromosome is composed of two blocks (copy one, and single-copy spacer) of equal length *L*. During a burn in phase (phase I), these blocks undergo mutation at rate *µ* and crossovers at rate *R*, where *R* is the number of events in the population in that block per generation. During phase II, a duplication event takes place in which the copy one block from a randomly chosen chromosome is copied to the right of the single-copy spacer. This duplication is conditioned to fixation following a neutral trajectory, after which, in phase III, neutral evolution with mutation and crossover occurs. During phases II and III, the original and duplicated blocks exchange information via IGC, which occurs at a rate *C* in all chromosomes carrying the duplication, where *C* is the number of IGC events in the population per generation.

We introduce five new features to SeDuS to better simulate the evolution of the *RPP8* gene family. First, we extend the simulator to model a three-copy gene family by including a second, unrelated spacer at the outset of the simulation, and by adding two additional phases to the simulation. During phase II, the “copy one” block is duplicated to the right of spacer one and left of spacer two. After phase III, a third duplicated block, copy three, is randomly selected, as in phase II, from the copy one block of a randomly chosen chromosome and introduced to the right of spacer two. In phase IV of the model, this duplication is conditioned to fixation following a neutral trajectory. In phase V, the neutral evolution of the five-block chromosome occurs, with mutation as in the original SeDuS and block-specific crossover rates, *R*. In phases IV and V, the copy one, two, and three blocks exchange information via IGC, which occurs at a total rate *C* in pairs of chromosomes carrying the duplication, which are selected randomly from the population to be the donor and acceptor of an IGC tract. 

Second, we allow the proportions of IGC between the three duplicated blocks to vary and add directionality in the amount of exchange between blocks. To do this, IGC occurs at a total rate *C*, and the frequency at which different pairs of copies of the gene family are chosen to be IGC tract donors and acceptors are independently specified as fractions of the total rate *C*. In our unequal exchange scenario, we summed the values of IGC estimated for *RPP8* paralogs to obtain *C* and divided each of the six estimates of IGC between pairs of *RPP8* paralogs by *C* to obtain a specific pairwise exchange fraction. This allowed copy three to undergo IGC with copy one at a higher rate than with copy two, as we observe at *RPP8*.

Third, we introduce non-tandem duplications by specifying block-specific rates of crossover, *R_C_* for each block representing copies one, two, and three, and *R_S_*_1_ and *R_S_*_2_ for single copy spacer blocks one and two. We then alter *R_S_*_1_ and *R_S_*_2_ relative to *R_C_*, allowing, for example, the specification of a tandem duplication by a *R_S_*_1_ equivalent to that of a 3 kb region in *A. thaliana* and an unlinked additional duplicated block with a *R_S_*_2_ equivalent to that of a 2.25 Mb region in *A. thaliana*, the same spacing as the *RPP8* gene family. 

Fourth, we introduce selfing as in [57], by specifying the fraction of reproductive events, *s*, in which a chromosome chooses to pair with itself or its sister chromosome, as opposed to picking another chromosome at random from the population. This involved the reduction of *C*, *R_S_*_1_, *R_S_*_2_, and *R_C_* by a factor of (1-*s*) to obtain “effective” IGC and crossover rates. 

Fifth, we introduce balancing selection at copy two for a presence–absence polymorphism by introducing an additional phase, phase VI, after the fixation of the three copies, in which copy two is constrained to have a frequency of 50%. This order of events was chosen due to the ancestral presence of three orthologs of *RPP8* in *A. lyrata*, and the subsequent creation of a CNV for D2 in *A. thaliana*. Selection in *RPP8* may not be balancing selection for a CNV, and instead may take the form of negative frequency-dependent selection based on *RPP8* functionality, or some other form. However, incorporating balancing selection for a CNV has characteristics similar to diversifying or negative-frequency-dependent selection, while still being possible to generate in this modeling framework. In phase VI, a randomly chosen chromosome loses its copy two block, and two haplotypes, one with all three copies present, and one with only copies one and three present, form a population with a CNV. This population is then simulated to evolve with selection for both haplotypes to be maintained at approximately 50% frequency. 

We used this extended version of SeDuS to vary individual parameters while holding other parameters constant at levels observed for *RPP8* in *A. thaliana*. The full sets of parameter values used in each simulation can be found in Appendix A. Generally, population size was held at *N* = 100, *R_C_* was held at 3.2 (equivalent to a 4 kb block, or four times the estimate of the population recombination rate, *ρ*, found for a 1 kb block found in [50]), *R_S_*_1_ was held at 2.4 (equivalent to a 3 kb block), *R_S_*_2_ was held at 1600 (functionally unlinked, equivalent to a 2 Mb block), *C* was held at 8.4 (the sum of the estimates for all six types of IGC at *RPP8*), *s* was held at 0.97 (the estimate of selfing in *A. thaliana* from [58], *µ* was held at 0.001, IGC tract length was the default, 100 bp, and the IGC exchange type was unequal exchange, using the relative frequencies estimated for *RPP8*. To explore the effects of specific parameters on nucleotide diversity in the gene family, three parameters were varied individually: paralog spacing, *R_S_*_2_, to simulate recombination in regions between 2 kb and 20 Mb in size; IGC rate, *C*, at 8 levels between 0.2 and 2000 events per generation; and exchange scenario, to consider both an equal exchange and the unequal exchange scenario observed for *RPP8* for IGC tract movement. We also considered the interaction between the fraction of selfing individuals, *s*, varied between 0 and 0.999, for three different values of *C*: 0.2, 8.4, and 200. A plot of example SeDuS outputs can be found in Appendix A.

We obtained equilibrium estimates for nucleotide diversity in phases III, V, and VI of the simulation by taking the average nucleotide diversity for 200 or more simulations for the generations where the majority of models had reached equilibrium. For phase III, this was between generations 4300 and 6500 and gave estimates of nucleotide diversity for two-copy gene families; for phase V, this was between generations 11,000 and 16,000 and gave estimates of nucleotide diversity for neutrally evolving three-copy gene families; and for phase VI, this was between generations 21,000 and 26,000 and gave similar estimates for three-copy gene families with copy two under balancing selection for a CNV. We divided these estimates by the average nucleotide diversity estimate for the single copy spacer one in generations 3000 through 15,000 to obtain the multiple of the single copy region reached by two and three copy gene families. 

## 3. Results

### 3.1. Variation in RPP8 Genomic Architecture, Phenotypes, and Sequence 

We genotyped 37 *A. thaliana* accessions for the presence or absence of each of the three paralogs of *RPP8*, then obtained sequence data for the entire paralogous region (4788 bp) for 31 *RPP8* paralogs: eight S, seven D1, seven D2, and nine P3 alleles (Appendix A), and sequence data for the LRR for 17 additional paralogs: three S, two D1, eleven D2, and one P3 alleles (Appendix A). These sample sizes were sufficient for reasonable estimation of population genetic parameters (Appendix A). We also obtained resistance data to the downy mildew pathogen *Hyaloperonospora arabidopsidis* for sixteen accessions, including twelve with full sequence data and four with LRR sequence data for S or D2 (Appendix A). We first describe gross features of individual *RPP8* paralogs: CNVs, within-gene insertions and deletions, and potential for functionality. We then discuss genotype-specific features, such as patterns of resistance to *H. arabidopsidis* and the evidence for particular crossover or IGC events between S and D1,D2 genotypes. Finally, we explore population genetic patterns of polymorphism and IGC. 

Our genotyped sample of 37 *A. thaliana* accessions yielded 14 single copy variants (S alleles) and 23 two-copy variants (D alleles D1 and D2) of *RPP8* (Figure 1b). The *RPP8* P3 locus was absent in one of 37 accessions, indicating copy number variation at this locus. S and D1,D2 CNVs were widely distributed throughout the species’ range, with both variants present in accessions of European, Asian, African, and North American origin. (Appendix A). D variants were more common in European samples, though they were not at a significantly higher frequency than the average CNV frequency (68% D1,D2; binomial *p* = 0.11) and S variants were more common in North America (12% D1,D2; binomial *p* = 0.013; Appendix A). To determine if CNV type or continent of origin were statistically dependent on the phylogenetic relationships between the alleles sampled, we determined the phylogenetic signal for both of these traits. CNV type had a significantly overdispersed phylogenetic signal for a sequence similarity tree of the LRR codons of *RPP8* (Bloomberg’s *K* = 0.504, *p* = 0.038, Appendix A), and a marginally significant phylogenetic signal for overdispersion on a tree of the non-LRR codons of *RPP8* (Bloomberg’s *K* = 0.389, *p* = 0.059). This indicated that alleles were more likely to be closely related to individuals of a different CNV class than alleles of the same CNV class. Continent of origin had no significant phylogenetic signal on a tree of either the LRR or the non-LRR codons of *RPP8* (Bloomberg’s *K* = 0.267; 0.291; *p* values = 0.69; 0.46), indicating that this sample evolved randomly with respect to continent of origin. Thus, this sample was suitable for obtaining a global view of the population genetics of *RPP8* paralogs (Appendix A). *A. lyrata* possesses the D1,D2 tandem duplication of *RPP8* and one copy of P3, indicating that the triplication of *RPP8* predated this speciation event. We did not find S variants in the small sample of *A. lyrata*.

We next examined the potential for functionality for the 31 alleles with complete sequence data. Within these alleles, Ler-0 had a nonsense mutation in P3, but no alleles had a frameshift mutation. Based on a comparison with *RPP8* orthologs in *A. lyrata*, there were six sites with amino acid insertion, and 20 sites with in-frame deletion. The majority of coding sequence insertions (4/6) were found in more than one sampled allele, while the majority of deletions were singletons (14/20). Within the introns, there were 32 different indels of varying lengths, 23 of which were found in more than one sampled allele. No indels were found near the borders of intron two, but P1 and P3 shared a 13 bp deletion directly adjacent to the 5′ border of intron one, and P3 samples also had 14 bp and 15 bp deletions at this border. These deletions did not compromise the GT-AG intron splicing motif for intron one, and thus we consider it unlikely that these samples retain and translate intron one as a result of this deletion. *RPP8* paralogs were on average 905 +/− 3 amino acids in length, with a range of 895–910 amino acids. Despite these differences in length, all *RPP8* paralogs retained most alignable homologous protein-coding residues and all but one had the potential for functionality in NLR gene-mediated recognition of pathogens.

We compared the *RPP8* genotype to phenotypic evidence for downy mildew resistance for sixteen accessions, including twelve that had complete *RPP8* paralogous sequence data and four that had LRR sequence data. Of the twelve, four were susceptible and eight were resistant; of the four with LRR data, three were susceptible and one resistant (Appendix A). There was no correlation between *RPP8* copy number and downy mildew resistance. Five of nine *RPP8* D variants and three of six *RPP8* S variants were resistant. The Cvi-0 variant without P3 was also resistant. There were no sequenced indels or amino acid variants within either the coding region or the LRR of S or D1 (the copy with allelic variation for downy mildew resistance/susceptibility), nor within either S, D1, or P3, that segregated perfectly with resistance; in fact, every site had at least two exceptions in correlations between single SNPs and resistance. Resistance and susceptibility had no phylogenetic signal on a tree of either the non-LRR and LRR region of *RPP8* (Bloomberg’s *K* = 0.247; 0.282, *p* = 0.89, 0.72), with almost every paralog from a susceptible plant most similar to a resistant *RPP8* allele (Appendix A). This phylogenetic distribution of resistance is consistent with multiple, independent losses (or gains) of downy mildew resistance in different alleles of *RPP8*. These resistance data are also inconsistent with a one SNP:one phenotype model of trait evolution [59].

We found a staggeringly large number of segregating sites—470 in total—in the full dataset, consisting of 31 fully sequenced alleles (17 accessions). This was more than four times more segregating sites than the majority of *NLR* genes in *A. thaliana* (Appendix A). We determined the number of fixed differences and frequencies of shared, derived polymorphism for each *RPP8* paralog to infer the propensity for allele-specific recombination in meiosis [42,60,61]. The number of segregating sites within paralogs ranges from 174 to 358 (Table 1). Intuitively, loci with a history of intergenic recombination should have few, if any, fixed differences, and derived allele frequencies should be positively correlated, with a linear relationship. *RPP8* S and D1 alleles had no fixed SNP differences and a large number (316) of shared derived polymorphisms. In contrast, S and D2 alleles had 27 fixed SNP differences (Table 2) and a smaller number (183) of shared derived polymorphisms (Table 3). Allele frequencies of shared polymorphisms were strongly correlated between S and D1 alleles (R^2^ = 0.70) and less correlated between S and D2 alleles (R^2^ = 0.14; Appendix A). These data indicate that S alleles pair (and exchange sequence via crossovers) predominantly with D1 alleles during meiosis and suggest that ectopic crossovers between S and D2 alleles are relatively infrequent. We thus treat S and D1 alleles as one homologous gene, P1, in subsequent sections, and D2 alleles as a paralogous gene (P2). 

Despite the clear homology of the *RPP8* paralog protein coding sequence, there was considerable pairwise genetic distance within and between paralogous *RPP8* alleles. At the nucleotide sequence level, there was an average of 5% pairwise coding sequence diversity. At the amino acid level, there was even higher coding sequence diversity: 9.14% pairwise diversity on average, equivalent to 82 (+/− 15) different amino acids between different alleles of *RPP8*. Surprisingly, there were no fixed SNP differences between D1 and P3 loci, and only four fixed differences between S and P3 loci (Table 2). S, D1, and P3 alleles had a similar number of shared polymorphisms (Table 3) and strongly correlated frequencies of shared polymorphisms (Appendix A). This indicates the presence of sufficient sequence exchange between the distant duplicates P1 and P3 to maintain homogenization of alleles. This exchange could not be mediated by ectopic crossing over, as this would lead to the loss of sequence between P1/P2 and P3 (or the loss of ~300 genes), which has never been observed. Instead, sequence homogenization between P1 and P3 must occur by IGC.

### 3.2. Rates of IGC between RPP8 Paralogs

IGC has been observed previously between P1 and P3 [46]. We tested for evidence of IGC between all three paralogs in three ways: linkage disequilibrium (LD), the allelic site frequency spectra (SFS), and phylogenetic inference (for reasons elaborated in the Methods). Using SFS, we also estimated *C*, or the number of sequence exchange or IGC events in the population per generation.

IGC reduces LD to an extent that increases with the rate of IGC in that region [55]. Recombination hotspots within a region undergoing IGC are also theorized to further reduce LD, but only in the hotspot region [55]. However, P1/P2 and P3 have lower than average recombination rates in *A. thaliana* [62]. This is likely due to the presence of a CNV in the mapped heterozygote, which could suppress crossover events [63,64]; alternatively, it could be due to the high nucleotide diversity at these loci, which has been observed to reduce crossover frequency in heterozygotes [65]. The pattern of LD within members of the *RPP8* gene family did not vary substantially within the sequenced region for any paralog and was thus not consistent with predictions of a recombination hotspot (Appendix A). In contrast, the LD measures, r^2^, were significantly less than the genome-wide expectation for the entire length of the sequenced region of P1 (Figure 1c). The average r^2^ between SNPs 1 kb apart in P1 was 0.078 (+/− 0.13), significantly less than the r^2^ of 0.52 measured genome wide for SNPs 1 kb apart in *A. thaliana* [50]. P2 also had reduced LD between SNPs 1 kb apart, 0.232 (+/− 0. 25), though LD at distances greater than 1 kb was not significantly different than that of the genome on average (Figure 1c). The average r^2^ between SNPs 1 kb apart in P3 was 0.167 (+/− 0.20), and r^2^ values were significantly less than genome-wide expectations for SNPs 2 kb apart or less (Figure 1c). After the duplication breakpoint 3′ of P3, r^2^ values became more typical of the genome-wide average (Appendix A), and r^2^ values were not atypical between D1 and D2, nor between P1/P2 and P3. Qualitatively, mild reductions in LD are theorized to result when *C* is close to one event per generation, and strong reductions in LD when *C* >> 1 [55]. The observed pattern of LD was consistent with a low rate of IGC between the other two paralogs and P2, and a high rate of IGC (*C* > 1) between P1 and P3.

SFS between pairs of loci undergoing IGC can be informative about the rate of sequence exchange, *C* [41]: as *C* increases, SFS are expected to have more shared SNPs and fewer (or no) differences between paralogs. To compare rates of IGC for paralogs with theoretical expectations, we constructed SFS between each pair of paralogs, which resulted in six SFS, or two SFS per pair. These SFS showed the frequencies of SNPs in the first, “acceptor” paralog that were either shared with the second, “donor” paralog or specific to the acceptor paralog (Appendix A). We randomly drew 1000 subsets of SNPs from the 31 fully sequenced paralogs to create bootstrapped distributions of observed SFS. We compared these spectra with expected spectra for duplicates undergoing gene conversion at three rates: *C* = 0.2, 1, and 5 [41]. The two SFS with P1 as the donor paralog were most similar to the expected SFS where *C* = 1 (95.5% and 98.7% of bootstrapped distributions; *p* < 2.2 × 10^−16^). The two SFS with P2 as the donor paralog were most similar to the expected SFS where *C* = 0.2 (85.5% and 97.7% of bootstrapped distributions; *p* < 2.2 × 10^−16^), while the SFS with P3 as the donor and P1 as the acceptor was most similar to the expected SFS where *C* = 5 (88.1% of bootstrapped distributions; *p* < 2 × 10^−16^). The SFS data were consistent with *C* of 1 or higher between P1 and P3, and with a lower rate, of 0.2 to 1, between P2 and the other paralogs (Figure 1d).

To better visualize and interpret IGC at *RPP8*, we created a novel three-copy SFS. This three-copy SFS included frequencies of derived SNPs specific to individual paralogs, or private SNPs; derived SNPs that were shared with only two of the three paralogs, and derived SNPs that were shared in all paralogs (Figure 1e). To determine if frequencies of shared and specific polymorphisms varied in different regions of the *RPP8* homologous sequence, we also plotted the frequencies of derived private and shared SNPs against position in the gene for all combinations of paralogs (Appendix A). In a three copy SFS, private derived SNPs at high frequencies indicate regions where IGC does not occur. No private SNPs, or SNPs unique to a single paralog, were fixed in the duplicated region of *RPP8* (Figure 1e and Appendix A). Most private SNPs were found at low frequencies: only 2% of private derived alleles were the major allele, and 86% of private SNPs occurred in just one or two of the sampled sequences. These data were consistent with the occurrence of IGC across the entire duplicated region of *RPP8*. We observed many SNPs shared between two paralogs that were not found in the third (Figure 1e and Appendix A). Just as an SFS indicates the relative frequency of IGC events between a single pair of paralogs, differences in the numbers of shared polymorphisms in three-copy polymorphism frequency plots indicate the relative frequency of IGC events between different pairs of paralogs. Most SNPs shared between two of the three paralogs were not shared with P2 (234 SNPs not shared with P2; 51 not shared with P3; 15 not shared with P1, Figure 1e). Very few SNPs were shared between P2 and P3 but not with P1, and these SNPs were found at low frequencies (Figure 1e). The data indicate that the most common type of IGC is between P1 and P3, then between P1 and P2, and least commonly between P2 and P3. 

Polymorphism is unevenly distributed across the *RPP8* locus. The third exon, which encodes the LRR, has approximately twice the density of the SNPs shared between the three paralogs than the non-LRR regions: 60% of the shared SNPs were found in the third exon, which is 35.8% of the length of the homologous region, and 40% of shared polymorphisms were found in the LRR, which is 20.7% of the length of the homologous region (Appendix A). In fact, all 31 fully sequenced alleles contained different haplotypes in the LRR region, reflecting the enormous allelic diversity and SNP polymorphism in this region of the gene, more than any singleton NLR gene (Table 1, Appendix A). Because the majority of shared polymorphism was located in the region surrounding the LRR region, we hypothesized that diversifying selection is maintaining distinct functional haplotypes in this domain.

To explore the potential for diversifying selection in the LRR versus the remainder of the protein, we compared two maximum parsimony trees, one constructed with hypothesized LRR codons from [7], and one with the non-LRR codons (Figure 2). These two trees revealed strikingly different branching patterns. In particular, the non-LRR portion from the three paralogous loci mapped to the tree consistent with independent evolution, with significantly more phylogenetic signal for paralog number than expected by chance (Bloomberg’s *K* = 1.073, *p* = 0.001, Appendix A). D1 and S alleles were interspersed on the tree, as expected for a single recombining locus (the P1 locus). There was little evidence of sequence exchange between P1 and P3: the P3 alleles formed a nearly monophyletic clade within the P1 alleles, except for one D1 allele (Figure 2a). The non-LRR branching pattern was consistent with a duplication event from S or D1 to create P3. As P3 shares its flanking sequence with the upstream region of D1, it was likely derived from this paralog before the split of *A. thaliana* and *A. lyrata*. There was no evidence for sequence exchange between P1 and P2; the P2 alleles formed a monophyletic outgroup to the remaining *RPP8* paralogs (Figure 2a). 

In contrast, the LRR tree supported an entirely different evolutionary history (Figure 2b). Here, there was less phylogenetic signal for paralog number than expected by chance (Bloomberg’s *K* = 0.325, *p* = 0.048, Appendix A), and S, D1, P2, and P3 alleles were all distributed paraphyletically across the tree, with low proportions of bootstrapped branch support for many deeper nodes in the consensus sequence tree. The largest clade was of four P3 alleles. The LRR tree supported frequent gene conversion events between P3 and P1 alleles: three of five clusters of two or more of P1 alleles had P3 alleles as their closest relative on the tree (Figure 2b). It also supported gene conversion events between D1 and P2 alleles: P2 alleles clustered into two smaller clades that each had D1 alleles as the clade’s closest outgroup (Figure 2b).

To further explore the allelic diversity and evolution in the LRR region, we sequenced an 888 bp region containing the 12 distal-most LRR repeats of *RPP8* for 17 additional alleles (Appendix A). Most alleles (45 of 50) had unique haplotypes in the LRR region (Table 1). A tree of these sequences also demonstrated high levels of paraphyly in alleles from different genomic locations (Appendix A). P3 alleles again had P1 alleles as their closest relatives, and P2 alleles had D1, S, and P3 alleles as their closest relatives. In addition, alleles from the same populations did not necessarily fall into the same clades or closely related clades. In three cases, we sequenced two or more P2 LRR from individuals isolated from the same population: two NFE-, three Pu-, and three Kz- alleles (as well as one NFE- and one Kz- allele from P1). Two pairs of alleles had identical haplotypes within the population; the remaining alleles had distinct haplotypes and pairwise nucleotide diversity of 5.1–6.9%, which was similar to between-population comparisons (0.0626 +/− 0.018). With the exception of the two pairs with identical haplotypes, all alleles from the same population were paraphyletic on the tree (Appendix A). These results indicated that similar levels of diversity and a huge number of distinct alleles have been maintained both within and between populations. 

### 3.3. Signatures of Selection in RPP8 Paralogs

In a predominantly selfing species such as *A. thaliana*, IGC between duplicates may be an important mechanism for moving new mutations onto different genetic backgrounds. In selfers, loci are typically homozygous, and given the age of the *RPP8* duplication event (originating prior to the split between *A. thaliana* and *A. lyrata*), the haplotypes at the paralogous loci should be distinct. With diversifying selection as a driving force, duplication and IGC may carry an additional selective benefit for selfers in both generating and spreading variation. To test this hypothesis, we looked for signatures of positive and balancing selection acting on *RPP8* paralogs in *A. thaliana*.

We explored the rate of *RPP8* protein evolution by comparing the K_a_:K_s_ ratio generated for comparisons of *A. thaliana* and *A. lyrata* alleles to the distribution of all genes shared between *A. thaliana* and *A. lyrata* [66,67]. Across the entire coding region, K_a_:K_s_ ratios within paralogs varied between 0.53 and 0.61 (Table 4), higher than 91–94% of the K_a_:K_s_ distribution in *A. thaliana*. The high K_a_:K_s_ ratio within *RPP8* was due to a large K_a_ relative to the genome-wide averages; K_s_ values were similar to the genome-wide average (Table 4). Within the LRR, K_a_:K_s_ ratios varied between 0.75 and 0.97 (Figure 3a and Appendix A), higher than 96–98% of the K_a_:K_s_ distribution. Amino acid diversity was highest within the 14 LRR subdomains of the LRR, particularly at the hypervariable “X” sites in the XXLXLXXXX LRR subdomain (Table 5). For these residues, the frequency of derived amino acid changes was 2.4- to 16-fold higher than the genomic background rate (Figure 3; Table 5). On average, 20–40% of “X” residues 2–6 in each LRR subdomain were derived, polymorphic residues, reflecting substitution rates 5- to 16-fold higher than typical. Nor were these derived amino acids simply single derived polymorphisms at high frequencies: “X” residues 2–6 had, on average, 2.8 to 3.5 amino acids segregating at each LRR subdomain (Table 5), with up to 7 amino acids segregating at some residues. This is indicative of positive selection for the retention of amino acid replacement changes in the LRR. 

If *RPP8* alleles are being maintained under balancing selection, positive values of Tajima’s D might be expected. However, duplicates undergoing IGC are theorized to have an underdispersed distribution of Tajima’s D compared to the single-copy gene case, with more than 95% of values within the region undergoing IGC falling between negative one and one in the neutral case, rather than between negative two and two [41]. As predicted by this work, a sliding window analysis of Tajima’s D found no regions with values above or below +/− 1.5 (Figure 3b and Appendix A). In total, 23% of windows in P2 had Tajima’s D’s above 1.0, however, with these windows all localizing within coding regions, and especially the LRR. In contrast, for P1 and P3 23% and 24% of 300 bp windows, again mainly in the LRR, had a Tajima’s D below −1.0. The data are suggestive of positive or purifying selection acting on the LRR of P1 and P3, and with balancing selection acting at P2.

### 3.4. Nucleotide Diversity

Gene duplicates undergoing IGC are theorized to have up to two times the level of synonymous nucleotide diversity (π_s_) in both copies relative to single copy genes [42]. However, values of π_s_ and nonsynonymous nucleotide diversity (π_a_) for the three paralogs were far higher than two times the single-copy expectations, even within the intron. π_s_ within and between the coding regions of the paralogs varied between 0.0333 and 0.0597, 6–12 times the genome average of ~0.005 (Table 4 and Table 5); and, 5.5–10 times the 5% tail of the distribution of π for a set of 800 single copy regions [67]. Nonsynonymous nucleotide diversity (π_a_) within the coding region of the paralogs was 20–27 times the genome average of ~0.0014 (Table 4). Ratios of π_a_/π_s_ were also higher than the genome average, particularly in the LRR (Figure 3a and Appendix A). A sliding window analysis of nucleotide diversity across each paralog showed that it varied between 2 and 15 times the genome average for P1, 0.5–12 times the genome average for P2, and 0.9–18 times the genome average for P3 (Figure 3c and Appendix A). The 700 bp intron sequence 5′ of the LRR had π_s_ levels 3.7, 2.7, and 4.6 times the genome average for P1, P2, and P3, respectively (Figure 3c and Appendix A). The LRR had π_s_ levels 10.3, 9.1, and 14.1 times the genome average, while the non-LRR coding region had π_s_ levels 6.1, 4, and 5.5 times the genome average, only slightly higher than the intronic region. Nucleotide diversity was also higher for *RPP8* than for the majority of single-copy NLR genes (Appendix A). Interestingly, no comparisons of π_s_ between paralogs were significantly different (Table 6). Synonymous divergence between *A. thaliana* and *A. lyrata* in the *RPP8* region was not in excess of genome-wide expectations, indicating that a high mutation rate was not responsible for increasing nucleotide diversity or π_s_ in these regions (Table 6). Instead, diversifying selection appears to be acting on all LRR subdomains within the LRR region.

### 3.5. Simulation of Neutral Evolution and Balancing Selection at RPP8

To simulate the effects of genomic features of the *RPP8* gene family on nucleotide diversity, we modified SeDuS, a two-locus forward-in-time simulator of population genetics in a duplicated gene family undergoing IGC [49], to accommodate three loci. We used SeDuS to vary the effect of paralog number, paralog spacing, IGC rate (*C*), IGC directionality, and selfing, all with and without selection to maintain a CNV, to determine which effects might lead to π_s_ levels 3–13 times the genome average, as seen for *RPP8*. Under neutrality, our simulation results confirmed our intuition that a three-copy system can have three times the nucleotide diversity of a one-copy system (Figure 4b). However, this level was only reached for three-copy gene families when *C* was unrealistically high (*C* ≥ 200). Variation in paralog spacing and IGC directionality had little effect on nucleotide diversity observed for each copy under neutral processes (Figure 4a,c). Under neutral processes, high levels of selfing led to a reduction in nucleotide diversity from three to two times the single copy average for intermediate IGC rates (Figure 5b), but minimally reduced nucleotide diversity from three times the single copy average for very high and very low levels of IGC (Figure 5a,c). No neutral scenario led to the level of nucleotide diversity seen in the LRRs of *RPP8*; the maximum nucleotide diversity seen was ~3.2 times the single copy average (Figure 5b,c).

We next simulated balancing selection at copy two, the simulated tandem duplicate linked to copy one, by introducing a copy number polymorphism at copy two maintained at 50% frequency in the population. Though the occurrence of IGC between paralogs of *RPP8* prevents a conventional interpretation of Tajima’s D with respect to balancing selection on D2, we were still interested in modeling the effects of this common CNV as if there was selection for its maintenance. We reasoned that, if the copy number variation at D2 was under balancing selection, then this should generate an increased level of polymorphism at P1, which is tightly linked to P2. A long-maintained reduction in the number of copies of P2 could also contribute to the halving of average diversity observed at P2 relative to the other paralogs (Table 6). Third, a copy number polymorphism could explain the asymmetry in the IGC rates we see between the paralogs: fewer copies of P2 should be reflected in IGC rates as a lower estimated rate of IGC with P2, which we observe in our IGC rate estimates (Figure 1d). 

Our simulation results showed that balancing selection acting on a CNV on P2 gave large increases in equilibrium nucleotide diversity for both the linked and distant copy of the gene family (Figure 4d–f). The distance between duplicates analogous to P2 and P3 had a minimal effect on nucleotide diversity for P1 and P3, and no effect at distances 200 kb and greater. When the IGC rate was less than 1, balancing selection increased nucleotide diversity at P1 to 12–13 times the level of single copy regions, depending on the type of IGC exchange. Intermediate levels of gene conversion gave the smallest increase in nucleotide diversity, while high levels of gene conversion (*C* ≥200) increased nucleotide diversity to 4–5 times the level seen for single copy regions, depending on the gene copy in question. The introduction of unequal rates of IGC and balancing selection did not influence the levels of nucleotide diversity at the three paralogs, except for a small increase in nucleotide diversity at copy one for low IGC rates (*C* = 0.2) (Figure 4f). Very high levels of selfing caused nucleotide diversity at both copy one and copy three to increase (Figure 5d–f). Selfing also led to a strong increase in nucleotide diversity in copy one when IGC rate was low (Figure 5d). We note that using parameters estimated for the *RPP8* gene family leads to a similar halving of nucleotide diversity in the copy with a CNV relative to the other two copies (3–4× vs. 2×), though the absolute level of nucleotide diversity was less than half that observed at *RPP8*.

## 4. Discussion

The evolutionary trajectory of duplicated genes is contingent on the frequency of sequence exchange between copies [43,61]. Just as speciation occurs more readily in allopatry, without gene flow [68,69], under neutral processes, gene duplicates that do not exchange sequence will quickly diverge from one another. Selection can likewise drive divergence among duplicates. In contrast, duplicates with even infrequent sequence exchange can become a factory for the generation of novel alleles. Duplicates with IGC will furthermore share more polymorphisms for a longer duration, as they evolve in concert [40,41,70]. Our study is a first attempt to investigate how these forces combine to influence the population genetics of an NLR gene.

We chose *RPP8* in *A. thaliana* to investigate the role of intergenic gene conversion on patterns and levels of genetic variation because the loci encoding this gene display at once three distinct forms of chromosomal duplication. First, one chromosomal type exists as a direct-repeat tandem duplication, with the two loci, D1 and D2, separated by only five kilobases. Exchange between the tandem copies in this chromosomal type is expected to be relatively frequent. A second chromosomal type has lost one copy of the gene, so that there is also a simple form of copy number variation at play. The single-copy allele, S, can exchange sequences with D1 and D2, either by intragenic crossing over or by gene conversion, when in a heterozygote carrying both chromosomal types. In *A. thaliana*, these heterozygotes are expected to be relatively rare, as the fraction of effective within-locus recombination events is ca. 3% of the total recombination rate [57,58]. Finally, there is a third copy of the gene, P3, located on the same chromosome, 2 Mb away from D1,D2 and S. A physical distance of this magnitude is not expected to present an obstacle for intergenic exchange with D1,D2 or S; the IGC rate is usually negatively correlated with the distance between paralogs and positively correlated with paralog sequence similarity [71,72,73]. A 2 Mb distance between loci gives rise to a reasonably high crossing over rate, approximately 2%: intergenic exchanges that create novel alleles within a chromosome can then be reshuffled across chromosomes at a reasonable frequency by crossing over. With these expectations, *RPP8* presented itself as a rich but tractable system to explore how duplicative and recombinational processes have influenced the variation, and potentially the evolution of a locus under strong selection.

We found that there is little shared variation between the tandem duplicates D1/S and D2 relative to the distant duplicates D1/S and P3, an unexpected finding that requires explanation. We can reject outright the possibility that intergenic exchange between duplicates in *RPP8* is rare, i.e., occurs at too low a frequency to prevent the divergence of the two loci. In fact, there is extensive sharing of SNP variation between D1/S and P3, though the two loci are separated by megabase, rather than kilobase distance. This would suggest that P1 and P2 likewise have the capacity for intergenic exchange that is not realized. Permanent heterozygote advantage between P1/P2 alleles is a plausible hypothesis for the selective advantage of differentiation between P1/P2 alleles; under this hypothesis, selection to maintain a permanent heterozygous configuration of alleles opposes intergenic exchanges that homogenize the two loci. This may be an example, therefore, of incipient, or even stable permanent heterozygote advantage with occasional intergenic exchange. 

Second, D1 and S are essentially indistinguishable from one another, with no fixed differences and 316 shared SNPs (Table 3). This is also an unexpected finding because direct exchange between these chromosomal types should be relatively rare; it requires intragenic recombination in an outcrossed heterozygote carrying both chromosomal types. The alleles at D1 and S also exhibit extensive reshuffling of SNPs, unlike the typical situation in *A. thaliana*, where loci tend to carry many fewer distinct haplotypes [57,74]. 

The major clue as to how this variation is shuffled between D1 and S comes from the additional observation that the physically distant locus, P3, shares the same constellation of SNPs (Figure 1e, Table 3). This is thus a three-locus gene-exchange circuit. Sufficiently frequent intrachromosomal gene exchange between D1 and P3 on the one chromosomal type, and between S and P3 on the other chromosomal type, coupled with sufficient crossing over between D1 or S and P3 in heterozygotes, can complete the circuit connecting the two paralogs on the two chromosomal types. This possibility is borne out by our simulations, as further elaborated below. 

A remaining feature of the genetic architecture requiring explanation is the lack of a duplicate on the chromosomal arm containing S. We posit that if alleles at P1 and P2 carry distinct specificities against pathogens, then perhaps P2 alleles (but not P1 alleles) exhibit a cost under certain conditions, such as in the absence of pathogens, as we previously found for *RPM1* and *RPS5* [32,33]. If so, the structural presence/absence polymorphism for P2 might be an adaptive chromosomal configuration in that deletions minimize the metabolic (or other) costs associated with expressing an NLR allele [32,33,75,76]. It is also possible that P2 and P3 have no canonical function in pathogen resistance, but instead exist to increase diversity at P1, the only paralog with mapped resistance specificities. However, given the extent to which variation is shared between P1 and P3, P3 may well share resistance functions with P1 and/or have generated novel resistance specificities. In principle, it should be possible to experimentally manipulate the configuration of alleles present at P1, P2, and P3 to test these hypotheses.

To summarize how this diversity is generated, while tandem duplicates P1 and P2 experience only a low level of intergenic exchange, perhaps due to selection to maintain permanent heterozygosity, extensive intergenic exchange between the D1/P3/S triad provides for an enlarged reservoir of variation compared to a single-copy locus, and the reshuffling of variation enables the continuous generation of diverse, novel alleles. 

The question then becomes, how is this variation maintained? Pathogen-mediated selection can lead to the maintenance of an extreme diversity of alleles, as seen at both plant and animal disease resistance loci [77]. There are three proposed mechanisms by which this occurs: heterozygote advantage, fluctuating selection and rare-allele advantage [78]. In single copy genes in a selfing species, heterozygote advantage cannot have an important role in increasing allelic diversity. We suggest that the unusual features of *RPP8* polymorphism result from the interaction between IGC and pathogen-mediated selection, and that this interaction has generated an allelic series that can confer recognition to multiple avirulence genes. The presence of three or more *RPP8* paralogs undergoing IGC makes the presence of multiple distinct copies of *RPP8* in an individual a virtual certainty. This creates a larger reservoir for the maintenance of variation within this gene family, and thus increases the age of alleles at the locus by allowing the persistence of older alleles in the reservoir.

Permanent heterozygosity by gene duplication restores the possibility that heterozygote advantage could select for diversity at *RPP8*. Most theoretical models which include heterozygote advantage as the only mechanism to generate polymorphism and consider realistic temporal distributions of fitnesses can maintain only two to eight unique alleles at a single locus [79,80]. Nevertheless, the extreme allelic diversity at *RPP8* is consistent with theoretical expectations for stable equilibria caused by heterozygote advantage alone, as an outcome when there is little difference in fitness among heterozygotes [79]. 

Multiple observations support selection on the LRR of at least one paralog to generate protein coding diversity at the population level. The K_a_:K_s_ ratio for each paralog is higher than 96–98% of the genome on average in the LRR, and slightly lower outside of the LRR, indicating diversifying selection acting on the LRR (Figure 3a and Appendix A). In addition, a huge variety of alleles are retained at each paralog: each fully sequenced allele is a unique haplotype, with levels of nucleotide diversity 9–14 times the genomic average within the LRR and 4–6 times the genomic average in the coding region outside of the LRR (Figure 3c and Appendix A), and in the tail of the distribution for both regions. At the hypervariable residues in each of the 14 LRR subdomains, the frequency of derived amino acids was 5–16 fold higher than the genomic average, strongly indicative of diversifying selection for novel functionalities (Table 6). An interaction between IGC and selection on the LRR is consistent with our observation that the proportion of shared polymorphism segregating at all three paralogs is greatest in the LRR (Appendix A), and with the highly paraphyletic distribution of allele genomic locations observed in the sequence similarity tree of the LRR rather than non-LRR codons (Figure 2). Note that IGC would not have to be concentrated on the LRR for such a pattern to occur; rather, diversifying selection could favor the maintenance of IGC events that overlap the LRR region, while the remainder of the gene could be evolving neutrally.

Under neutrality, the reservoir of variation maintained by multiple *RPP8* paralogs on multiple chromosomal types undergoing IGC was predicted to be larger than that of a single copy gene. We wanted to determine whether the depth of this reservoir was sufficient to account for the levels of polymorphism we observed at *RPP8*. We therefore conducted neutral forward-in-time simulations of the evolution of a three-copy gene family with different IGC and selfing rates. Neutral simulations with three copies of the gene family were close to the levels of nucleotide diversity seen in the introns of *RPP8* paralogs, but were lower than levels in coding regions, even with the unrealistically high rates of IGC necessary to maximize nucleotide diversity under neutrality (Figure 4). 

In an effort to achieve higher levels of diversity, we simulated balancing selection for maintenance of a CNV at P2. We used this type of selection to simplify the modeling but note that the effects of balancing selection, driven by costs, and frequency-dependent selection acting on resistance specificities, would be equivalent. Balancing selection was effective in increasing nucleotide diversity at both P1 and P3. The increase in nucleotide diversity was minimized for intermediate rates of IGC and was most similar to the levels of nucleotide diversity seen in the non-LRR coding sequence rather than the LRR coding sequence. Low rates of IGC and high rates of selfing both increased nucleotide diversity at P1 to levels seen at *RPP8* but failed to increase nucleotide diversity at P2 and P3 to that seen for the gene family (Figure 5d). Regardless, our simulations indicate that balancing selection acting at one locus can increase diversity at both linked and unlinked genes that are undergoing IGC. 

Our simulations find that the distance between loci has almost no effect on the level of polymorphism created in such a system, meaning that both tandemly duplicated clusters and distant duplicates may be undergoing similar processes that lead to increased diversity. However, we note that this model does not take into account known effects of chromosomal interference at distances shorter than 200 kb. Chromosomal interference is known to reduce crossovers and the number of double-strand breaks, thus also reducing the number of noncrossover events [81]. Regardless, the 2 Mb distance between the *RPP8* paralogs is an order of magnitude greater than 200 kb. The opposite orientation and large distance between the paralogs might allow IGC to occur by chromosomal looping to form interhomolog joint molecules during recombination, similar to how homeologous chromosomes interact in polyploid species.

NLR genes in many plant species are present in large, tandem arrays, which are frequently located on the ends of chromosomes, far from the pericentromeric suppression of recombination [15,16,18,20,21,22,23,24]. Ectopic crossovers are often invoked as a likely mechanism to generate NLR gene diversity in these clusters [16,20,82]. However, evidence for ectopic crossover events involving gene gain or loss has thus far rarely been observed in studies of NLR gene polymorphism [83,84,85,86], and crossovers are known to be suppressed in such regions [63,64]. In addition, for the singleton NLR genes *RAC1* and *RPP13*, strong negative relationships have been observed between nucleotide polymorphism in heterozygous NLR genes and crossover frequency [65]. Crossover inhibition, which can lead to a preponderance of non-crossover events such as IGC, is thus likely to be particularly strong near complex NLR gene families, due to both structural and nucleotide polymorphism in these regions. IGC can also generate novel haplotypes and has been previously reported between NLR gene paralogs [46,87]. In conclusion, we propose that IGC may be a common feature for NLR genes across plant species under diversifying selection and in gene clusters.

## Figures and Tables

**Figure 1 genes-10-00691-f001:**
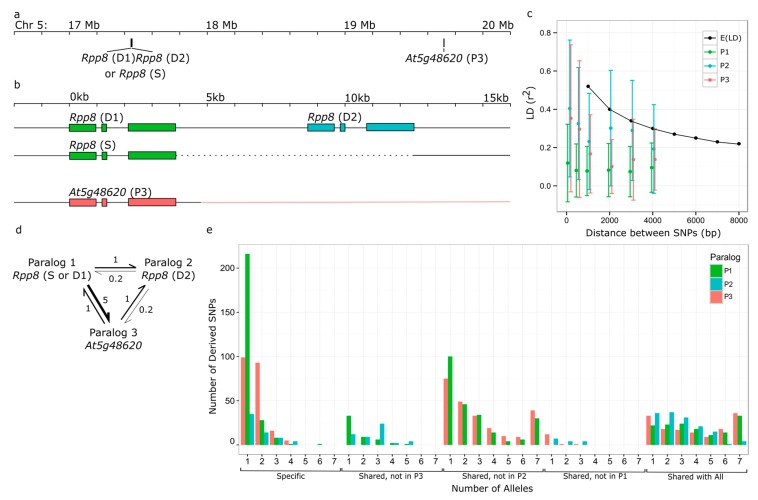
Signatures of intergenic gene conversion (IGC) in the *RPP8* gene family in *Arabidopsis thaliana*. *RPP8* refers to the entire gene family, including all three paralogs. D1,D2 and S refer to specific chromosomal types. P1, P2, and P3 refer to specific paralogs, and green, blue, and orange represent P1, P2, and P3, respectively. (**a**) Positions of the coding sequence of *RPP8* and *At5g48620* (P3) on chromosome 5. (**b**) Homology of loci in the *RPP8* gene family. Positions are shown in kilobases, relative to the start codon of the first paralog at that chromosomal location. Exons of *RPP8* loci are shown as boxes; dashed line indicates deleted regions; orange line indicates regions of P3 with no paralogy to P1 or P2. (**c**) Expected and observed linkage disequilibrium (LD) decay from [50] and in three *RPP8* paralogs. (**d**) Schematic of IGC rate estimates and directionality between *RPP8* paralogs most consistent with the SFS and LD results from (**c**,**e**), and Appendix A. (**e**) Modified site frequency spectrum (SFS) between all three *RPP8* paralogs. Plot shows the frequencies of derived SNPs specific to a paralog or shared with one or both other paralogs.

**Figure 2 genes-10-00691-f002:**
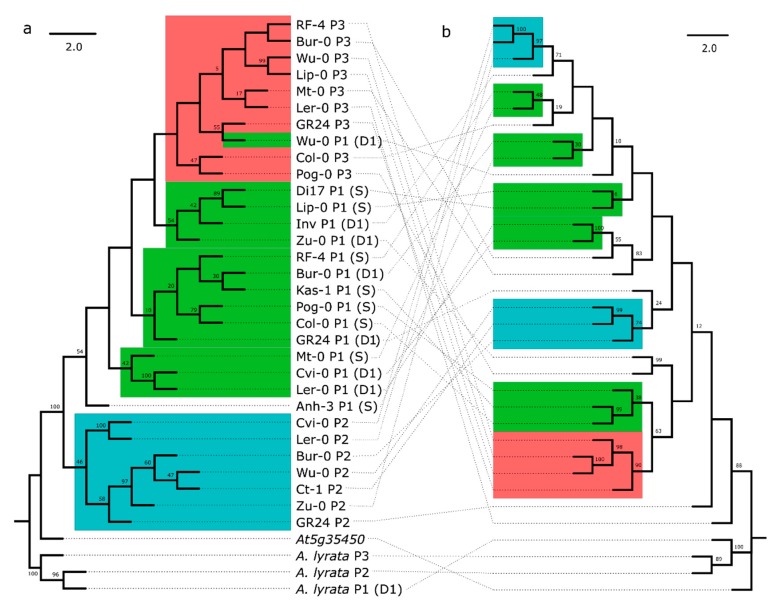
Bootstrap consensus trees for the maximum parsimony phylogenies of the leucine-rich repeat region (LRR) and non-LRR regions of the three *RPP8* paralogs, P1, P2, and P3. Clades comprised of alleles from one paralog are boxed. Green, blue, and orange boxes represent P1, P2, and P3, respectively. The percentage of replicate trees in which the associated taxa clustered together in the bootstrap test (5000 replicates) are shown next to the branches; values less than 5 are not shown. (**a**) Sequence similarity tree of the non-LRR region. 239 out of 1701 sites were parsimony informative. (**b**) Sequence similarity tree of the framed LRR region for the same accessions as in (**a**). In total, 236 out of 1019 sites were parsimony informative.

**Figure 3 genes-10-00691-f003:**
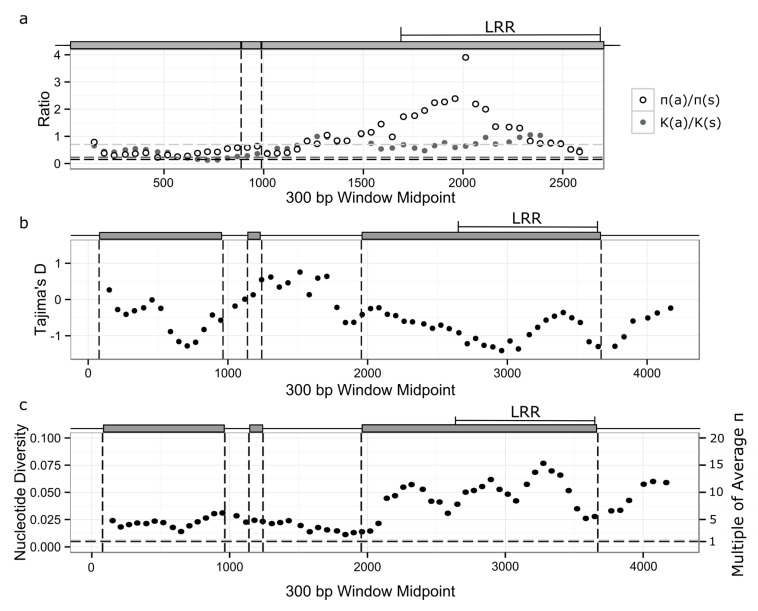
Sliding windows of within-species polymorphism and divergence between *Arabidopsis thaliana* and *A. lyrata* for paralog P1 at *RPP8*. Plots for paralogs P2 and P3 can be found in Appendix A. Grey boxes above the plots represent positions of exons of P1. Vertical lines indicate exon boundaries, as shown in the schematic above each plot. The leucine-rich repeat region (LRR) is also indicated. (**a**) π_a_:π_s_ and K_a_:K_s_ within *A. thaliana* and between *A. thaliana* and *A. lyrata.* Black and dark grey dashed horizontal lines indicate average levels of π_a_:π_s_ and K_a_:K_s_ within *A. thaliana* and between *A. thaliana* and *A. lyrata*; light grey dashed line is the 95% right-hand tail for K_a_:K_s_. (**b**) Tajima’s D. (**c**) Synonymous nucleotide diversity. The black horizontal dashed line indicates the average level of nucleotide diversity within *A. thaliana*; the line width is the confidence interval for average nucleotide diversity.

**Figure 4 genes-10-00691-f004:**
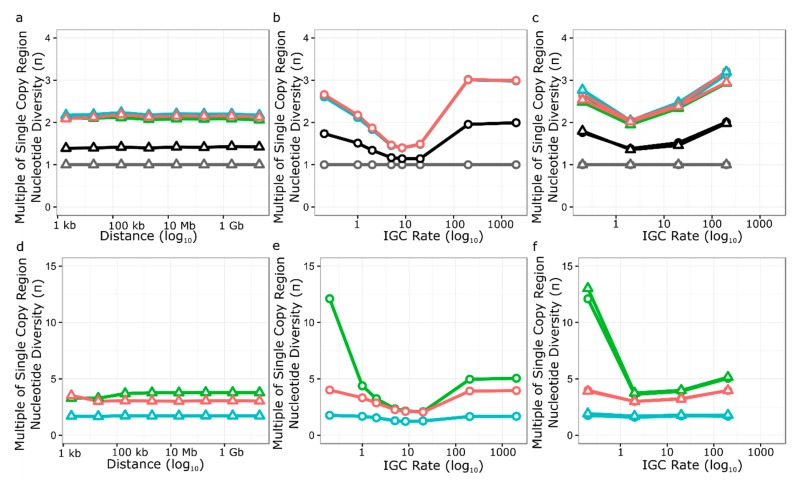
Simulation results for two and three copy gene families undergoing intergenic gene conversion with varying distance between copy two and copy three and varying IGC rates. Each point represents the average equilibrium nucleotide diversity from 200 independent extended SeDuS simulations. Nucleotide diversities relative to a single copy locus for one copy (grey), two copy (black), and three copy (green, blue, and orange are copies one, two, and three, respectively) systems are shown. Open circles represent simulations with equal exchange between copies, while open triangles represent simulations with unequal exchange between the three copies. (**a**–**c**) Neutrality. (**d**–**f**) Balancing selection maintaining copy two at 50% frequency within the population. (**a**,**d**) Varying the distance between copy two and copy three. (**b**,**e**) Varying the total IGC rate. (**c**,**f**) Varying the total IGC rate and the type of IGC tract exchange.

**Figure 5 genes-10-00691-f005:**
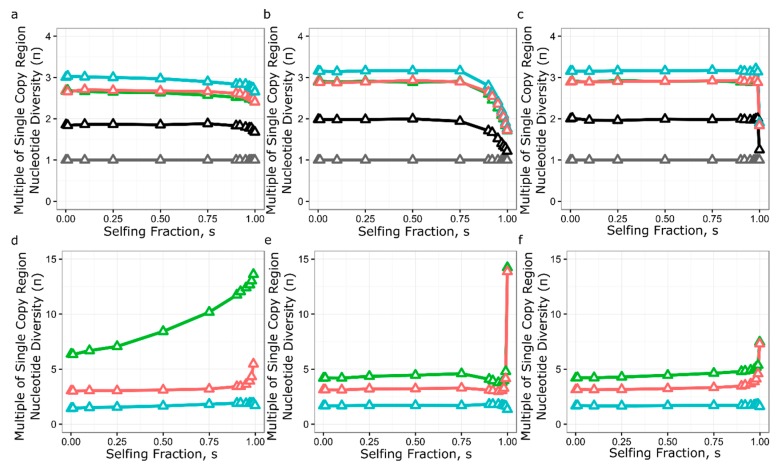
Effect of selfing fractions and IGC rates for two and three copy gene families undergoing intergenic gene conversion. Each point represents the average equilibrium nucleotide diversity from 200 independent extended SeDuS simulations. Nucleotide diversities relative to a single copy locus for one copy (grey), two copy (black), and three copy (green, blue, and orange are copies one, two, and three, respectively) systems are shown. (**a**–**c**) Neutrality. (**d**–**f**) Balancing selection maintaining copy two at 50% frequency within the population. (**a**,**d**) IGC rate = 0.2. (**b**,**e**) IGC rate = 8.4. (**c**,**f**) IGC rate = 200.

**Table 1 genes-10-00691-t001:** Polymorphism within the coding and leucine-rich repeat region (LRR) of *RPP8* paralogs.

	All Coding Sites (2766 bp)			LRR (888 bp)		
	Segregating Sites (S) ^1^	Average Number of SNPs/Allele ^1^	Number of Unique Haplotypes	Segregating Sites (S) ^1^	Average Number of SNPs/Allele ^1^	Number of Unique Haplotypes
P1	358	108.0	15/15	254	61.2	20/21
S	276	109.3	8/8	153	54.2	11/11
D1	246	103.4	7/7	212	68.0	9/10
D2	174	80.5	7/7	162	51.8	17/19
P3	302	110.8	9/9	179	63.7	9/10
all	470	118.0	31/31	327	60.4	45/50

^1^ Values were determined excluding gaps in pairwise comparisons only.

**Table 2 genes-10-00691-t002:** Fixed derived SNPs in the sequenced duplicated region at locus X, rows, not segregating at locus Y, columns.

In X, Not in Y ^1^	S	D1	D2	P3
S	-	0	27	0
D1	0	-	14	0
D2	4	0	-	3
P3	4	0	27	-

^1^ There are 470 segregating sites when all 31 sequenced paralogs of the *RPP8* gene family are included.

**Table 3 genes-10-00691-t003:** Number of shared derived polymorphisms in the sequenced duplicated region between locus X, rows, and Y, columns.

In X and Y ^1^	S	D1	D2	P3
S	-	316	183	326
D1	-	-	161	307
D2	-	-	-	162

^1^ There are 470 segregating sites when all 31 sequenced paralogs of the *RPP8* gene family are included.

**Table 4 genes-10-00691-t004:** Within-paralog synonymous and nonsynonymous nucleotide diversity (π) and divergence (K) for the entire coding region and the leucine-rich repeat region (LRR).

	Coding Region						LRR	
	π_s_	π_a_	π_a/_π_s_	K_s_	K_a_	K_a_/K_s_	π_a/_π_s_	K_a_/K_s_
Genome Average ^1^	0.005 (0.004–0.006)	0.0014	0.23	0.13 (0.02, 0.24)	0.025 (0, 0.12)	0.19 (0, 0.07)	n/a	n/a
P1	0.0429	0.0355	0.829	0.143	0.0789	0.527	1.56	0.746
P2	0.0341	0.0279	0.815	0.130	0.0823	0.612	2.27	0.965
P3	0.0459	0.0383	0.830	0.144	0.0814	0.538	1.06	0.751

^1^ Average values for these variables for all coding regions in the genome are shown in this row, and 95% confidence intervals, where available, are shown in parentheses.

**Table 5 genes-10-00691-t005:** Coding sequence diversity in the hypervariable (“X”) sites of the 14 leucine-rich repeat (LRR) subdomains in 30 alleles in the *RPP8* gene family. Table shows the number of derived amino acids, typically out of 30, present at each amino acid residue in each LRR of all 30 fully sequenced *RPP8* paralogs.

LRR ^a^	1 ^b^	2	3	4 ^c^	5^d^	6	7	8 ^e^	9	10 ^f^	11	12	13	14	Average Fraction Derived (~K_a_) ^g^	Fold Increase over Genomic K_a_ (C.I.) ^h^	Distinct Amino Acids (C.I.) ^i^
l	0	0	0	0	0	0	0	0	0	3	0	0	0	0	0.007 +/− 0.002	0.3	1.1 (1, 1.5)
x	0	0	0	0	0	0	0	2	0	0	0	4	0	0	0.014 +/− 0.003	0.6	1.2 (1, 2)
x	0	0	0	0	0	0	3	1	1	1	0	0	0	0	0.014 +/− 0.002	0.6	1.2 (1, 2)
l	0	0	0	0	0	0	0	0	0	7	0	0	1	0	0.019 +/− 0.004	0.8	1.1 (1, 2)
x	0	0	0	0	6	0	4	0	0	0	4	14	0	0	0.067 +/− 0.01	2.7	1.4 (1, 2)
x	0	0	0	1	7	0	1	0	1	0	0	0	0	0	0.024 +/− 0.004	1	1.3 (1, 2)
l	0	0	1	0	0	0	0	0	0	0	0	0	0	0	0.002 +/− 0.001	0.1	1.1 (1, 1.5)
X	21	0	0	0	0	1	2	0	0	0	0	0	1	0	0.06 +/− 0.013	2.4	1.4 (1, 2)
X	11	1	1	4	1	7	26	5	12	16	0	1	1	0	0.205 * +/− 0.018	8.2*	2.9* (1.5, 4.5)
L	8	0	0	1	2	0	2	0	1	6	0	1	0	0	0.05 +/− 0.006	2	1.6 (1, 2.5)
X	25	4	10	10	9	22	17	28	2	3	13	22	0	3	0.4 * +/− 0.022	16*	3.5* (2, 5.5)
L	2	0	0	4	5	6	1	18	1	21	1	0	0	0	0.14 * +/− 0.016	5.6*	1.9 (1, 3)
X	25	2	12	12	0	0	3	7	11	3	18	0	8	12	0.269 * +/− 0.018	10.8*	2.9 (1, 5)
X	0	3	25	3	11	0	0	6	3	18	18	10	2	5	0.248 * +/− 0.019	9.9*	3.3* (1.5, 5.5)
X	21	0	0	10	18	0	1	7	1	18	18	13	0	0	0.255 * +/− 0.02	10.2*	2.8 (1, 5)
X	4	0	11	6	2	0	0	18	0	0	0	6	14	0	0.145 * +/− 0.014	5.8*	1.9 (0.5, 3.5)
x	0	0	1	9	19	2	0	9	0	15	0	0	0	0	0.131 +/− 0.015	5.2	1.7 (1, 3)
x	0	0	0	8	0	12	4	11	0	0	7	0	0	1	0.102 +/− 0.011	4.1	1.6 (1, 3)
x	0	0	0	7	0	0	1	10	0	3	0	0	0	0	0.05 +/− 0.007	2	1.6 (0.5, 3)
x	0	0	0	2	3	2	1	7	0	2	13	3	0	0	0.079 +/− 0.009	3.2	1.6 (1.5, 2.5)
x	0	0	0	0	0	0	4	3	0	0	0	0	0	1	0.019 +/− 0.003	0.8	1.2 (1, 2)
x	0	0	0	0	1	0	1	3		0	1	0	0	0	0.015 +/− 0.002	0.6	1.3 (1, 2)
x		0	0	0	1		0	18		0	0	0	0	0	0.058 +/− 0.016	2.3	1.2 (1, 2)
x		8		0			1	3		0	0	0	1	0	0.048 +/− 0.01	1.9	1.4 (1, 2)
x		0		0			0	1		0	0			1	0.01 +/− 0.002	0.4	1.1 (1, 2)
x							0	0						1	0.011 +/− 0.006	0.4	1.3 (1, 2)

* Values where the confidence interval did not overlap the confidence interval for genome. ^a^ Bold, capitalized X and L represent the LRR subdomain, which encompasses the putative β strand/β turn region involved in pathogen recognition. X and *x* represent any site. ^b^ This LRR is proposed as an update to McDowell and Dangl 1998. It fits the LRR motif criteria and better matches the pattern at the other 13 LRR subdomains. In addition, one (*n* = 7) indel was not included. ^c^ One rare (*n* = 1) indel was not included in the analysis of all 14 LRRs. ^d^ The ancestral state reconstruction was changed from a stop codon to K for one amino acid in this LRR. ^e^ One rare (*n* = 1) indel was not included in the analysis of all 14 LRRs. ^f^ One site in this LRR was not included in the analysis of all 14 LRRs, because it was unique to this LRR. ^g^ The average fraction of nonsynonymous amino acids (K_a_) at that site across all LRRs. ^h^ The fold increase in the fraction of nonsynonymous amino acid changes relative to the typical genomic nonsynonymous amino acid changes (K_a_), and the 95% confidence interval for that value. ^i^ The average number of amino acids segregating at that site across all 14 LRRs, and the 95% confidence interval for that value, rounded to the nearest 0.5 amino acids.

**Table 6 genes-10-00691-t006:** Synonymous nucleotide diversity (π_s_) in the coding region between locus X, rows, and Y, columns, and the standard deviation in π_s_.

X/Y	S	D1	D2	P3
S	0.0477 ± 0.0100	0.0449 ± 0.0079	0.0537 ± 0.0067	0.0526 ± 0.0130
D1	-	0.0392 ± 0.0110	0.0559 ± 0.0076	0.0476 ± 0.0138
D2	-	-	0.0333 ± 0.0156	0.0597 ± 0.0140
P3	-	-	-	0.0451 ± 0.0195

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
