# Peer review of "Population Genetics of the Highly Polymorphic RPP8 Gene Family"

_genes, 2019, doi:10.3390/genes10090691_

Round 1

Reviewer 1 Report

MacQueen et al. present a very nice study on the three-locus RPP8 system in A. thaliana. I was impressed by their meticulous analysis and the explanations given in intricate detail. In particular I like to comment on the statistical rigour with which results were analysed.

Only a few minor points need to be addressed in my opinion.

p5 The use of parsimony should be better justified and explained. p6 Is A. lyrata a good outgroup given the estimated age of the system (preceeding the split, as per p9). Since I guess that no other/better outgroup is available this should be explained and discussed. Figure 2 could be improved by flipping the tree in part b to face the tree in a. The authors could also connect branches of both trees with dashed lines. This would illustrate the jumping of the branches even better. p13 line539ff This actually makes me wonder if the tree topology could be an artefact of the parsimony reconstruction methods. The authors might consider running Bayesian phylogeny to double-check their results.

Kind regards

Dr. Philipp Schiffer

Author Response

MacQueen et al. present a very nice study on the three-locus RPP8 system in A. thaliana. I was impressed by their meticulous analysis and the explanations given in intricate detail. In particular I like to comment on the statistical rigour with which results were analysed.

Only a few minor points need to be addressed in my opinion.

p5 The use of parsimony should be better justified and explained.

Edited, p. 5 lines 212-218.

p6 Is A. lyrata a good outgroup given the estimated age of the system (preceeding the split, as per p9). Since I guess that no other/better outgroup is available this should be explained and discussed.

We agree and elaborate on why the frequent sequence exchange makes A. lyrata a reasonable outgroup. p. 6 lines 229-232.

Figure 2 could be improved by flipping the tree in part b to face the tree in a. The authors could also connect branches of both trees with dashed lines. This would illustrate the jumping of the branches even better.

We agree and have made this edit.

p13 line539ff This actually makes me wonder if the tree topology could be an artefact of the parsimony reconstruction methods. The authors might consider running Bayesian phylogeny to double-check their results.

Our reasoning for including the similarity trees wasn't to support any particular phylogeny, but to demonstrate that the LRR & non-LRR reconstructions were dramatically different. We ran a maximum likelihood method to double-check this result. We used a JTT matrix-based model and a discrete Gamma distribution to model evolutionary rate differences among sites (5 categories +G, parameter = 0.3263). For the non-LRR, 30.83% sites were allowed to be invariable; for the LRR, 3.95% sites were allowed to be invariable. When nodes had even moderate bootstrap support, they were not affected by the choice of maximum parsimony or maximum likelihood. Both reconstruction methods lead to the same interpretation - substantial sequence exchange in the LRR of Rpp8, and much less sequence exchange in the non-LRR region of Rpp8.

Kind regards

Dr. Philipp Schiffer

Reviewer 2 Report

Review: Genes: Population Genetics of highly polymorphic RPP8 gene family.

This is a interesting paper. A lot of works have speculated about mechanisms that maintain or generate diversity of NLR loci, but extensive, sequence-based testing as in this paper has not really been done. That is to say, not to this extend for one specific locus. Overall, the paper seems very comprehensive. The methods are well described as are the results.
I don't have any major points.

There are a few smaller points which I indicate below.
In most cases the authors do a very good job explaining there analyses, but in some cases, some additional explanations might be useful. Genes is a relatively broad journal and many researchers with an interest in NLRs are not familiar with the interpretations of the population genetics analyses, but will be interested in it. A final check to see whether all terminology and data ainterpretation is well explained at the point of presentation could be beneficial to these readers.

The introduction and discussion are well written. One criticism that one could have is that the intro, as well as the discussion are very much arabidopsis focused. To me as a non-arabidopsis researcher, this begs the question how much of these findings are relevant to other species as well. Some specific non-arabidopsis references are cited (e.g. 15, 18, 21 and 23) but they're al grouped together under one statement of NLR clutsers. There's a nice amount of work out on NLR clusters in solanaceous spp (potato, tomato, pepper) e.g. Seo et al 2016 Frontiers in Plant Science) or in rice (Stein et al 2018 Nat genetics) Highlighting such commonatilties between arabidopsis and other crops might make this paper more appealing to a broader audience. It might also be interesting to point out that in our study of NLR diversity within one inbreeding wild tomato population (Solanum pennellii, LA0716), we found Rpp8 and Rpp13 homologs among the few genes that showed large diversity and possible signs for balancing selection. (Stam et at GBE 2016).

The sequencing and methods for data analysis seem solid. I am not familiar with SeDuS, so I cannot fully comment on the validity of the methods, but seeing the overal high quality of the rest of the manuscript. If not already done / planned to do so, I would recommend to share the modified code on a repository like github or zenedo so that others can utilise it.

l402 staggeringly large. Really? Are there similarly detailed data available on another locus in several arabidopsis accessions? It would really help to put the reported numbers (470 sites) into context.

l460 C is described as %, L461,462 C is described by number w/o % indication >> 1 and > 1.

l470, which dataset was used to draw these snps?

For Figure S6 takes a while to take in and to spot the differences between the legends and understand the data. Maybe it helps to make associations with the main text if the term donor (as used on lines 472, 473, 475) was also included in the graphs and the legend.

Figure S6 and l472-476. The authors state that dependent on the donor SFS are most similar to an expected SFS simulated with different values of C. The simulations with other values of C are however not shown. To convincingly make this point it would be useful to see these simulation. Especially since the simulated values in e.g. panel a and h show a mich higher Fixed allele count and also the singleton allele frequeny appears to be off in panels b,d,e,g.

Figure S7, would is be possible to label the genes themselves with names as well. E.g. add P1, P2 and P3 above each plot. I keep forgetting which color represents which.

l505 what is the origin of the data that you compared with?

l527, without stating numbers it's hard to interpret / validate sentences like "most P3 alleles had P1 alleles as neighbor". It is not clear how one would count this.

Figure S9, It is hard to quickly see the different paralogs of the populations. It would be good to include them and to color code for example the names in the tree. E.g. all Kz P2s in green, all Pu P2s in red. This will help to illustrate the point of within population diversity.

l565 confusingly phrased. I'd say: Ka's were similar to the genome-wide average.

l796, I think mentioning whether this is outside the 95% CI or tail of any expected distribution is a better indicator than comparing with the (known to be low) genomic average

All figures / legends: the boxes are often referred to as Red boxes, whereas the dots are referred to as orange. On my screen, these all look like the same color.

Author Response

This is an interesting paper. A lot of works have speculated about mechanisms that maintain or generate diversity of NLR loci, but extensive, sequence-based testing as in this paper has not really been done. That is to say, not to this extend for one specific locus. Overall, the paper seems very comprehensive. The methods are well described as are the results.

I don't have any major points.

There are a few smaller points which I indicate below.

In most cases the authors do a very good job explaining their analyses, but in some cases, some additional explanations might be useful. Genes is a relatively broad journal and many researchers with an interest in NLRs are not familiar with the interpretations of the population genetics analyses, but will be interested in it. A final check to see whether all terminology and data ainterpretation is well explained at the point of presentation could be beneficial to these readers.

The introduction and discussion are well written. One criticism that one could have is that the intro, as well as the discussion are very much arabidopsis focused. To me as a non-arabidopsis researcher, this begs the question how much of these findings are relevant to other species as well. Some specific non-arabidopsis references are cited (e.g. 15, 18, 21 and 23) but they're al grouped together under one statement of NLR clutsers. There's a nice amount of work out on NLR clusters in solanaceous spp (potato, tomato, pepper) e.g. Seo et al 2016 Frontiers in Plant Science) or in rice (Stein et al 2018 Nat genetics) Highlighting such commonalities between arabidopsis and other crops might make this paper more appealing to a broader audience. It might also be interesting to point out that in our study of NLR diversity within one inbreeding wild tomato population (Solanum pennellii, LA0716), we found Rpp8 and Rpp13 homologs among the few genes that showed large diversity and possible signs for balancing selection. (Stam et at GBE 2016).

We add these references and edit the introductory paragraphs to emphasize that similar features for NLR genes have been found in many plant species. We then narrow our focus to touch on specifics of what has been found in A. thaliana. We emphasize this again in the discussion (lines 63,65, 69, 72, 857).

The sequencing and methods for data analysis seem solid. I am not familiar with SeDuS, so I cannot fully comment on the validity of the methods, but seeing the overall high quality of the rest of the manuscript. If not already done / planned to do so, I would recommend to share the modified code on a repository like github or zenedo so that others can utilise it.

We can make the code available by request and will work with the coauthor who extended the software to see if he will be willing to share this on a public repository.

l402 staggeringly large. Really? Are there similarly detailed data available on another locus in several arabidopsis accessions? It would really help to put the reported numbers (470 sites) into context.

We agree and added text emphasizing that this number is large even for NLR genes (lines 413-414).

l460 C is described as %, L461,462 C is described by number w/o % indication >> 1 and > 1.

Line 452 edited: C is the number of events per generation

l470, which dataset was used to draw these snps?

Edited, "from the fully sequenced paralogs". (line 480)

For Figure S6 takes a while to take in and to spot the differences between the legends and understand the data. Maybe it helps to make associations with the main text if the term donor (as used on lines 472, 473, 475) was also included in the graphs and the legend.

Edited - we now include this in the graphs and the legend.

Figure S6 and l472-476. The authors state that dependent on the donor SFS are most similar to an expected SFS simulated with different values of C. The simulations with other values of C are however not shown. To convincingly make this point it would be useful to see these simulation. Especially since the simulated values in e.g. panel a and h show a mich higher Fixed allele count and also the singleton allele frequency appears to be off in panels b,d,e,g.

We agree these SFS don't perfectly match the expectations - in part, we suspect this is because these are expectations for a two-paralog system. All three expected distributions that we compared the simulated distributions to are displayed in this plot, however, in panels a, b, and d. We now emphasize this in the caption.

Figure S7, would is be possible to label the genes themselves with names as well. E.g. add P1, P2 and P3 above each plot. I keep forgetting which color represents which.

Edited - names added.

l505 what is the origin of the data that you compared with?

The data we compared with was Sanger sequencing on ~1kb regions of the LRR for 56 to 92 accessions and for 27 singleton NLR genes from Bakker et al 2006. We have this information in the caption (Figure S8).

l527, without stating numbers it's hard to interpret / validate sentences like "most P3 alleles had P1 alleles as neighbor". It is not clear how one would count this.

Edited to clarify this - thanks.

Figure S9, It is hard to quickly see the different paralogs of the populations. It would be good to include them and to color code for example the names in the tree. E.g. all Kz P2s in green, all Pu P2s in red. This will help to illustrate the point of within population diversity.

Agreed - Pu now has grey boxes, and Kz has black boxes.

l565 confusingly phrased. I'd say: Ka's were similar to the genome-wide average.

Edited.

l796, I think mentioning whether this is outside the 95% CI or tail of any expected distribution is a better indicator than comparing with the (known to be low) genomic average

Edited - to in the tail of the distribution.

All figures / legends: the boxes are often referred to as Red boxes, whereas the dots are referred to as orange. On my screen, these all look like the same color.

Agreed - we've changed these to say orange, except for Figure S8, which is unambiguous (has only red).